# Somatostatin Receptor 2 Expression Profiles and Their Correlation with the Efficacy of Somatostatin Analogues in Gastrointestinal Neuroendocrine Tumors

**DOI:** 10.3390/cancers14030775

**Published:** 2022-02-02

**Authors:** Hirofumi Watanabe, Fumiyoshi Fujishima, Izumi Komoto, Masayuki Imamura, Susumu Hijioka, Kazuo Hara, Yasushi Yatabe, Atsushi Kudo, Toshihiko Masui, Takahiro Tsuchikawa, Kazuhiro Sakamoto, Hisashi Shiga, Tomohiro Nakamura, Naoki Nakaya, Fuyuhiko Motoi, Michiaki Unno, Hironobu Sasano

**Affiliations:** 1Department of Pathology, Graduate School of Medicine, Tohoku University, Sendai 980-8575, Japan; rowingnet0991@gmail.com (H.W.); ffujishima@patholo2.med.tohoku.ac.jp (F.F.); 2Department of Surgery, Kansai Electric Power Hospital, Osaka 553-0003, Japan; i-komoto@mx.bw.dream.jp (I.K.); imamura.masayuki@c4.kepco.co.jp (M.I.); 3Kansai Electric Power Medical Research Institute, Osaka 553-0003, Japan; 4Department of Hepatobiliary and Pancreatic Oncology, National Cancer Center, Tokyo 104-0045, Japan; rizasusu@gmail.com; 5Department of Gastroenterology, Aichi Cancer Center Hospital, Nagoya 464-0021, Japan; khara@aichi-cc.jp; 6Department of Pathology and Clinical Laboratories, National Cancer Center, Tokyo 104-0045, Japan; yyatabe@ncc.go.jp; 7Department of Hepato-Biliary-Pancreatic Surgery, Graduate School of Medicine, Tokyo Medical and Dental University, Tokyo 104-0045, Japan; kudomsrg@gmail.com; 8Department of Surgery, Kyoto University Graduate School of Medicine, Kyoto 606-8507, Japan; tmasui@kuhp.kyoto-u.ac.jp; 9Department of Gastroenterological Surgery II, Hokkaido University Graduate School of Medicine, Sapporo 060-8638, Japan; tsuchi0108@gmail.com; 10Department of Pathology, Osaki Citizen Hospital, Osaki 989-6183, Japan; ksakamoto-och@h-osaki.jp; 11Department of Gastroenterology, Tohoku University Graduate School of Medicine, Sendai 980-8574, Japan; shiga@med.tohoku.ac.jp; 12Department of Health Record Informatics, Tohoku Medical Megabank Organization, Tohoku University, Sendai 980-8573, Japan; tnakamura@megabank.tohoku.ac.jp; 13Department of Preventive Medicine and Epidemiology, Tohoku Medical Megabank Organization, Tohoku University, Sendai 980-8573, Japan; naoki.nakaya.c2@tohoku.ac.jp; 14Department of Surgery I, Yamagata University Graduate School of Medical Science, Yamagata 990-2331, Japan; fmotoi@med.id.yamagata-u.ac.jp; 15Department of Surgery, Graduate School of Medicine, Tohoku University, Sendai 980-8575, Japan; m_unno@surg.med.tohoku.ac.jp

**Keywords:** neuroendocrine tumor, somatostatin receptor 2, foregut NET, hindgut NET, immunohistochemistry, digital image analysis

## Abstract

**Simple Summary:**

Immunolocalization of somatostatin receptor 2 (SSTR2) could predict the therapeutic efficacy of somatostatin analogues (SSAs) in neuroendocrine tumors (NETs). Therefore, in this study, we evaluated SSTR2 immunoreactivity and elucidated its correlation with clinicopathological variables, including the therapeutic response to SSAs in gastrointestinal neuroendocrine tumors (GI-NETs) using digital image analysis (DIA) and other established methods of evaluation. SSTR2 immunoreactivity in foregut NETs was significantly higher than that in hindgut NETs. SSTR2 immunoreactivity was significantly negatively correlated with the Ki-67 labeling index in foregut NETs but positively correlated in hindgut NETs. Therefore, the significance of SSTR2 immunoreactivity in GI-NETs is considered to be different according to the primary sites. We also first demonstrated that DIA could provide a good alternative for predicting response to SSAs in evaluating SSTR2 immunoreactivity of GI-NETs.

**Abstract:**

Somatostatin analogues (SSAs) are widely used to treat gastroenteropancreatic neuroendocrine tumors (GEP-NETs). Somatostatin receptor 2 (SSTR2) immunoreactivity serves as a predictive marker of the therapeutic efficacy of SSAs in pancreatic NETs. However, SSTR2 expression profiles in tumor cells and their association with the therapeutic efficacy of SSAs remains virtually unknown in gastrointestinal NETs (GI-NETs). Therefore, we evaluated the association between SSTR2 immunoreactivity and embryological origin and proliferative activity in 132 resected surgical tissues of GI-NETs. The correlation between SSAs’ therapeutic efficacy and SSTR2 immunoreactivity was evaluated in 14 GI-NETs treated with SSAs. SSTR2 immunoreactivity was evaluated using Volante scores, immunoreactive scores, and digital image analysis (DIA). SSTR2 immunoreactivity was significantly negatively and positively correlated with the Ki-67 labeling index in foregut and hindgut NETs, respectively. In the normal mucosa, neuroendocrine cells in the rectum had significantly lower positive rates of SSTR2 than those in the stomach and duodenum. SSTR2 expression profiles in GI-NETs could differ by primary sites, while the difference of those between foregut and hindgut NETs might be derived from the SSTR2 status of normal neuroendocrine cell counterparts. In addition, DIA could provide a good alternative for predicting response to SSAs in evaluating SSTR2 immunoreactivity of GI-NETs.

## 1. Introduction

The incidence of gastrointestinal neuroendocrine tumors (GI-NETs) has recently increased, possibly because of an increased administration of proton pump inhibitors and developments in detection equipment such as endoscopy [1,2,3]. GI-NETs are classified as foregut, midgut, and hindgut NETs according to their embryological origins and their clinicopathological features [4,5,6]. GI-NETs are both clinically and pathologically heterogeneous tumors [7,8,9]. This heterogeneity is also pronounced in their endocrine activities and tumor growth, possibly because GI-NETs originate from a variety of neuroendocrine cells located in different sites [10]. Almost all GI-NETs have been reported to express somatostatin receptors (SSTRs), and their expression profiles have also been known to be heterogeneous [7,8,11].

Somatostatin, a peptide hormone that binds to the SSTR, has been reported to be expressed in many parts of the body, including the pituitary glands, pancreas, and gastrointestinal tract [12,13,14]. Somatostatin was reported to be involved not only in the inhibition of hormonal activities but also in cell proliferation [9,12]. Somatostatin analogues (SSAs), including octreotide and lanreotide, have been widely used as therapeutic agents in gastroenteropancreatic neuroendocrine tumors (GEP-NETs) due to their anti-secretory and/or anti-proliferative activities [9,15].

SSTRs are composed of five different subtypes of G-protein-coupled transmembrane receptors (SSTR1–5) [12,15,16,17]. The SSTR2 status in NET tumor cells evaluated by immunohistochemistry has been reported to be associated with the therapeutic efficacy of SSAs in pancreatic NETs (PanNETs) [18,19] and acromegaly [20]. Recently, peptide receptor radionuclide therapy (PRRT) has been employed as a new therapy of NETs, and a correlation with SSTR2 status of NETs has been also suggested [21,22]. However, the association between SSTR2 expression profiles in tumor cells and the therapeutic efficacy of SSAs remains virtually unknown in GI-NETs, especially in foregut and hindgut NETs. Moreover, the detailed correlation between SSTR2 status and the proliferative activities of GI-NETs according to embryological classification has also remained unexplored at this juncture.

Various scoring systems of SSTR2 immunoreactivity based on eyeball analysis, such as Volante score [18,19] and Immunoreactivity Score [11,20], have been proposed in order to yield a more reproducible and subjective interpretation of results toward establishing a better correlation between SSTR2 immunohistochemistry results and SSAs’ therapeutic efficacy. However, these attempts have been mostly reported in PanNETs, not in GI-NETs. In addition, those analyses all had inevitable and/or intrinsic limitations, including inter-and intra-observer variability [23,24]. For example, when evaluating membranous SSTR2 immunoreactivity, it is challenging to determine whether the immunoreactivity obtained represents the complete or circumferential membrane based on manual and eyeball analysis [24]. Therefore, various attempts have been proposed to develop more objective methods, including digital image analysis (DIA) [24].

Therefore, in this study, we attempted to explore the followings in GI-NETs: 1) the possible association between SSTR2 immunoreactivity and the embryological origin and the proliferative activity of tumor cells, 2) the correlation between the therapeutic efficacy of SSAs and SSTR2 immunoreactivity in GI-NET patients treated with SSAs, with particular emphasis on immunoreactivity evaluated by DIA, and 3) SSTR2 immunoreactivity in the normal neuroendocrine cell counterparts from which tumor cells originate in order to account for the diversity of SSTR2 immunoreactivity in tumor cells.

## 2. Materials and Methods

### 2.1. Tissues and Patient Characteristics

In total, 145 GI-NET tissue specimens (Appendix A) obtained from 141 patients (Appendix A) were retrieved retrospectively from the surgical pathology files at Hokkaido University Hospital, Tohoku University Hospital (Miyagi, Japan), Osaki Citizen Hospital (Miyagi, Japan), Aichi Prefectural Cancer Center Hospital (Nagoya, Japan), Noe Hospital (Osaka, Japan), Tokyo Medical and Dental University Hospital, Kansai Electric Power Hospital, and Kyoto University Hospital. We tentatively excluded the patients receiving chemotherapy before surgery and those with metastatic lesions. After applying the exclusion criteria following a careful review of patients’ charts, 132 cases of resected surgical tissues turned out to be available for examination (Appendix A, Group 2). Fourteen GI-NET cases treated with SSAs, octreotide or lanreotide, were also available for determining SSA therapeutic efficacy in order to explore the association between SSAs’ therapeutic efficacy and SSTR2 immunoreactivity (Appendix A, Group 3). SSAs’ therapeutic efficacy was determined according to the Response Evaluation Criteria in Solid Tumors (RECIST) [25]. When multiple tissue samples were available from a single patient, the tumor tissue sample that was therapy-naïve, collected shortly after SSA administration, or from the primary site was selected [18]. Complete response (CR) was detected in one patient, stable disease (SD) in nine patients, and progressive disease (PD) in four patients. To further explore the correlation between SSAs’ therapeutic efficacy and SSTR2 immunoreactivity, the patients were tentatively classified as “CR or SD” and “PD” in this study. Moreover, all GI-NET cases were categorized as foregut, midgut, or hindgut NETs according to their primary locations (gastric or duodenal NETs were considered foregut NETs, jejunal or ileal NETs were considered midgut NETs, and rectal NETs were considered hindgut NETs; [Appendix A]) [7], and as grade (G) 1, G2, or G3 according to histological grade based on World Health Organization (WHO) classification [26,27]. The research protocol of this particular study was approved by the IRB or Ethics committee of the institutions above.

Furthermore, 37 resected surgical tissue specimens (Group 4), including those from the non-neoplastic mucosa of the stomach, duodenum, and rectum from 32 patients who received surgery at Tohoku University Hospital from 2019–2020, were available for the evaluation of SSTR2 immunoreactivity of non-pathological neuroendocrine cells in corresponding tissues of the gastrointestinal tract. The inclusion criteria of the non-neoplastic mucosal tissues in this study were summarized as follows: (1) patients with no previous histories of NETs, endocrine diseases including diabetes mellitus, and inflammatory bowel diseases, (2) patients with no past histories of chemotherapy, and (3) the locations of the non-neoplastic mucosa tissue were different from the location of the tumors or previously resected sites. A total of 19 men and 13 women were included in this group. Three tissues included only gastric mucosa, nine included only duodenal mucosa, five included both gastric and duodenal mucosa, and fifteen included only rectal mucosa.

Serial tissue sections of 10% formalin-fixed paraffin-embedded (FFPE) tissues were prepared in this study. Two of the authors (HW and HS) independently reviewed the specimens and confirmed the histopathological diagnoses.

### 2.2. Single Immunohistochemistry

One representative tissue section containing the tumor with the greatest dimension was selected for each case, and corresponding serial tissue sections (3–4-μm thick) were prepared. The detailed protocols of immunohistochemistry used in this study are summarized in Table 1. All immunostained slides were scanned digitally using the Nanozoomer S360 (C13220-01, Hamamatsu Photonics, Shizuoka, Japan) for subsequent image analyses. 

### 2.3. Double-Immunohistochemistry

Double immunohistochemical staining was performed using chromogranin A and SSTR2 in 32 Group 4 patients comprising non-neoplastic mucosa to study the immunoreactivity of SSTR2 only in neuroendocrine cells. SSTR2 immunostaining was performed using a single immunostaining method, and the antigen–antibody complexes were visualized using diaminobenzidine (DAB) (brown).

Antigen retrieval was subsequently performed using 10 mM citrate buffer (pH 6.0) by heating in a microwave at 210 W for 15 min. The slides were then washed with phosphate buffer saline (PBS) and incubated with protein blocking solution (Histofine Kit, Nichirei) at room temperature. The slides were then reacted with the primary antibody (chromogranin A) overnight at 4 °C and incubated with a secondary antibody and alkaline phosphatase-conjugated streptavidin (Nichirei). The antigen–antibody complex of chromogranin A was visualized in red color using fast-red (fast-red IIsubstrate kit; Nichirei) and counterstained with hematoxylin.

### 2.4. Evaluation of Ki-67 Labeling Index

The Ki-67 labeling index (LI) was evaluated according to the counting method defined by the WHO in 2019 with some modifications [26,27] using HALO^®^ CytoNuclear v1.6 (Indica Laboratories, Corrales, NM, USA) as previously reported [5,24] and was rounded down to two decimal places.

### 2.5. Evaluation of Single Immunohistochemistry of SSTR2 Immunoreactivity Using Manual/Eyeball Analysis (Volante Scores and Immunoreactive Scores) and DIA

SSTR2 membranous immunoreactivity was evaluated using a quantitative or semi-quantitative method according to three different scoring systems: Volante scores [19], immunoreactive scores (IRSs) [20], and scoring using DIA [24]. The Volante scores and IRSs were both obtained independently by two of the authors (HW and HS). When disagreement occurred in the eyeball-based methods between these two observers above, consensus was obtained by simultaneous observation using a dual-headed light microscope with subsequent discussion. These scores were evaluated in the same manner as reported in previous studies [19,20]. Volante scores ranged from 0 to 3 and were evaluated by localization of the immunoreactivity (cytoplasm or cell membrane), the extent of membranous immunoreactivity (circumferential or incomplete), and positivity of the positively stained cells (more or less than 50%) [18,19]. IRSs ranged from 1 to 12 and were calculated by multiplying the positivity of the positively stained cells (0: 0%, 1: <10%, 2: 10–50%, 3: 51–80%, or 4: >80%) and the staining intensity (0: none, 1: weak, 2: moderate, or 3: strong staining) [20]. The representative illustrations of the SSTR2 immunoreactivity are summarized in Figure 1a–d.

DIA was performed using the HALO^®^ Membrane v1.7 (Indica Laboratories, Corrales, NM, USA) [28]. After reviewing the entire tumor areas, the “hot spot” areas, which were annotated digitally, were tentatively selected. The “hot spot” areas demonstrated the most marked SSTR2 immunoreactivity, including 1000–2000 tumor cells. In this study, we avoided evaluating insufficiently fixed areas. When evaluating the results of the tumor cells, parameters such as “Nuclear size” and “Nuclear Roundness” were defined according to the method reported in a previous study [5]. For the recognition and scoring of SSTR2 immunoreactivity, the membrane stain optical densities of the DAB reaction product (RGB 0.268,0.570,0.776) for SSTR2 immunoreactivity were unified in all the cases examined in this study [28]. When detecting membranous immunopositive cells, the parameter of “Minimum Membrane Completeness” was set. This parameter was defined as the minimum completeness of the membrane, with the cell count being a percentage of the membrane immunoreactivity. This enabled the quantitative evaluation of the cell [24]. For example, when setting the parameter as 50%, tumor cells with membranous circumferential completeness under 50% were excluded from the calculation, and when setting the parameter as 0%, tumor cells with membranous circumferential completeness over 0% were included in the calculation (Appendix A) [24]. In order to further calculate the absolute values of SSTR2 immunoreactivity regardless of the positivity of the membranous circumferential completeness, we set the parameter as 0%. After setting these parameters, the data reflecting SSTR2 immunoreactivity, such as “% Positive cells” (percentage of positive cells), “Avg Positive Cell Membrane OD” (average membrane stain optical densities for the DAB reaction product of positive cells), and “Avg Positive Membrane Completeness” (average completeness or circumferential features of positive immunoreactivity), were automatically determined [24]. We calculated the quantitative SSTR2 immunoreactivity using DIA according to the following formula: “% Positive cells” × “Avg Positive Cell Membrane OD” x “Avg Positive Membrane Completeness.”

### 2.6. Evaluation of Double Immunohistochemical Staining of SSTR2 Immunoreactivity

All double-stained slides were evaluated using eyeball analysis. After reviewing the entire epithelial areas with care, we determined the “hot spot” areas, which contained the largest number of double positive-SSTR2 and chromogranin A cells, counting at least 10 chromogranin-positive cells in one or two high power fields (field number: 25). We then calculated the SSTR2 positive rates in the neuroendocrine cells as follows: “the number of SSTR2 and chromogranin A double-positive cells/number of chromogranin A-positive cells.” In the stomach, the epithelia with intestinal metaplasia were tentatively excluded from this evaluation. The difference between the two groups, the stomach and rectum, duodenum and rectum, were subsequently statistically evaluated. Representative illustrations of SSTR2 and chromogranin A double-positive cells are presented in Figure 1e,f.

### 2.7. Statistical Analyses

The correlations between Volante scores, IRSs, and SSTR2 immunoreactivities evaluated using DIA were analyzed using Wilcoxon/Kruskal–Wallis test and Spearman’s tests. A correlation coefficient (ρ) was obtained by using Spearman’s test. Differences of age, sex, histological grades or Volante scores, IRSs, and SSTR2 immunoreactivities evaluated using DIA among the GI-NETs, and the positive rate of SSTR2 of neuroendocrine cells of non-neoplastic mucosa were analyzed using the χ2 test or Wilcoxon/Kruskal–Wallis test. The significance level of the Wilcoxon test performed on the difference of the positive rate was corrected using Bonferroni’s inequality. The correlations between the Ki-67 LI, histological grades, and SSTR2 immunoreactivity according to the various scores were analyzed using Spearman’s tests and χ2 tests or Wilcoxon/Kruskal–Wallis tests, respectively. The concordance between the therapeutic efficacy of SSAs and the Volante scores, IRSs, and SSTR2 immunoreactivity evaluated by DIA were examined using receiver operating characteristic (ROC) curves, with calculations of the areas under the curves (AUCs), sensitivity, and specificity. AUCs were evaluated by the DeLong test. The significance level of the DeLong test was corrected using Bonferroni’s inequality. The statistical significance was set at *p* < 0.05. All the statistical analyses except for the DeLong test were performed using the JMP Pro ver. 16.0.0 (SAS Institute, Cary, NC, USA). The DeLong test was performed using R version 3.6.2 (R Foundation, Vienna, Austria).

## 3. Results

### 3.1. The Correlations of SSTR2 Immunoreactivity Evaluated by Three Scoring Systems

The correlations of SSTR2 immunoreactivity were evaluated using the three scoring systems in the cases of Group 2 or Group 3. Significant correlations were detected between the SSTR2 immunoreactivity evaluated using all three scoring systems (see Appendix A: *p* < 0.0001; *p* < 0.0001, ρ = 0.5972; and *p* < 0.0001, ρ = 0.7549, respectively).

### 3.2. SSTR2 Immunoreactivity Evaluated by Three Scoring Systems and Their Correlations with Embryological Origins, Ki-67 LI, and Histological Grades in Group 2

Concerning the Volante scores, 31 out of 56, 3 out of 5, and 10 out of 71 cases yielded a score of 3 in the foregut, midgut, and hindgut NETs, respectively (Table 2). When all the GI-NET (Appendix A) and NET Grade 1 (G1) cases (Figure 2a–c) were evaluated according to the scoring systems above, foregut NETs had significantly higher immunoreactivity than hindgut NETs (*p* < 0.0001). Concerning the GI-NET Grade 2 (G2), foregut NETs showed higher immunoreactivity than hindgut NETs using IRSs and DIA (*p* = 0.0522, *p* = 0.0611, respectively; Figure 2d–f). Regarding the GI-NET Grade 3 (G3), there were no significant differences in SSTR2 immunoreactivity between the foregut and hindgut NETs evaluated by all three scoring systems used (Figure 2g–i). The number of midgut NET cases was too small to be compared statistically with foregut or hindgut NETs.

In all the cases examined, no significant correlations were detected between the SSTR2 immunoreactivity evaluated using all the scoring systems and Ki-67 LI (Appendix A). Furthermore, the SSTR2 immunoreactivity evaluated by all three scoring systems was not significantly different among the NET G1, G2, and G3 cases (Appendix A). However, SSTR2 immunoreactivity was significantly inversely correlated with Ki-67 LI (Figure 3a–c) in the foregut NETs, especially when using the Volante scores and IRSs (Figure 3a,b, *p* = 0.0049, ρ = -0.3709 and *p* = 0.0099, ρ = -0.3418, respectively). SSTR2 immunoreactivity was not significantly different among G1, G2, and G3 cases (Appendix A), but the results obtained by the IRSs were close to being significantly different (Appendix A, *p* = 0.0631). In midgut NETs, SSTR2 immunoreactivity tended to be positively correlated with Ki-67 LI, although not significant (Figure 3d–f). SSTR2 immunoreactivity was also not significantly different between G1 and G2 (Appendix A). In the hindgut NETs, SSTR2 immunoreactivity was positively correlated with Ki-67 LI (Figure 3g–i), especially when evaluated using the Volante scores and IRSs; the first correlation was significant (Figure 3g,h, *p* = 0.0044, ρ = 0.3339 and *p* = 0.0600, ρ = 0.2244). SSTR2 immunoreactivity was also significantly different among G1, G2, and G3 cases when using the Volante scores (Appendix A, *p* = 0.0388), but not significantly different when using IRSs and DIA (Appendix A).

### 3.3. SSTR2 Immunoreactivity and Its Correlation with the Therapeutic Efficacy of Somatostatin Analogues in Group 3 When Evaluated Using Three Scoring Systems 

The clinicopathological characteristics of the Group 3 patients are summarized in Table 3. SSAs’ therapeutic efficacy was compared to the SSTR2 immunoreactivity evaluated by Volante scores, IRSs, and DIA with ROC curve analyses (Figure 4a–c). The AUC was highest when using DIA (AUC, 0.65; cut-off, 898.945; sensitivity, 80%; specificity, 75%), compared with Volante scores (AUC, 0.6125; cut-off, 2; sensitivity, 90%; specificity, 25%) and IRSs (AUC, 0.5875; cut-off, 4; sensitivity, 70%; specificity, 50%). No statistically significant differences were detected in those AUCs (Appendix A). SSTR2 immunoreactivity scores greater than the cut-off value were tentatively regarded as positive, and those with less than the cut-off value were considered negative. For example, a Volante score of two or three was regarded as positive, and 0 or 1 as negative. Higher proportions of SD or CR were achieved in positive cases compared to negative ones regardless of the scoring systems employed (Figure 4d–f). Moreover, SSTR2 immunoreactivity evaluated by DIA was most closely associated with SSAs’ therapeutic efficacy (Figure 4f, *p* = 0.0949), compared to Volante scores and IRSs (Figure 4d,e).

### 3.4. SSTR2-Positive Rates in Normal Neuroendocrine Cells of the Gastrointestinal Mucosa in Group 4

The tissue origins and clinical characteristics of the Group 4 patients and SSTR2-positive rates are summarized in Table 4. The SSTR2-positive rates in the stomach and duodenum were significantly higher than in the rectum (Figure 5, *p* = 0.0003, *p* < 0.0001, respectively).

## 4. Discussion

In this study, we first evaluated SSTR2 immunoreactivity in GI-NETs using both manual methods, including Volante scores [19] and IRSs [20] and DIA [24]. Furthermore, we also explored the association between the findings obtained by those analyses above and the primary sites of the GI tract, Ki-67 LI, histological grades, and SSAs’ therapeutic efficacy.

This study’s results demonstrated that SSTR2 immunoreactivity in foregut NETs was significantly higher than that in hindgut NETs, especially in NET G1, G2 cases (Figure 2). Moreover, SSTR2 immunoreactivity was significantly negatively correlated with Ki-67 LI in foregut NETs but positively in hindgut NETs (Figure 3). These results first demonstrated that SSTR2 expression profiles in GI-NETs could be different according to their primary sites. In addition, in the normal mucosa, neuroendocrine cells in the rectum showed less SSTR2 expression than those in the stomach and duodenum (Figure 5). In the normal gastric neuroendocrine cells, SSTR2 was reported to be present in enterochromaffin-like (ECL) cells [29], which is consistent with this study’s results. Therefore, based on those findings above, we hypothesized that the difference in SSTR2 profiles between foregut NETs and hindgut NETs was derived from the SSTR2 status of normal neuroendocrine cell counterparts (Figure 6). Low grade foregut NETs, mostly derived from their normal counterparts [30] with abundant SSTR2 expresssion, harbor relatively higher SSTR2 expression (Figure 6, left blue arrow). However, as the tumor progressed to higher histological grades, possibly in conjunction with genetic/epigenetic alterations such as the SSTR2 promoter hypermethylation [31,32], foregut NETs could be more deviated from the normal counterparts (Figure 6, right blue arrow) and demonstrate lower SSTR2 expression. In contrast, hindgut NETs derived from their normal counterparts, which had intrinsically low SSTR2 expression. Therefore, low grade hindgut NETs harboring relatively lower SSTR2 expression could result from neoplastic transformation of those normal counterparts (Figure 6, right red arrow). With tumor progression to higher histological grades, hindgut NETs could be more deviated from their normal counterparts and harbor higher SSTR2 expression (Figure 6, left red arrow). However, hindgut NETs in intermediate grades also yielded lower SSTR2 expression than foregut NETs (Figure 6). Further investigations are warranted to clarify this interesting hypothesis. In addition, it could be interesting to explore the differences of gene profiles between foregut and hindgut NETs resulting in that difference of SSTR2 expression profiles in future investigations.

Various histopathological scoring systems have been proposed to evaluate SSTR2 immunoreactivity to predict the potential therapeutic efficacy of SSAs, especially in PanNETs [19,20,33]. In our present study, we initially demonstrated that the SSTR2 immunoreactivity evaluated using the scoring systems above could serve as potential predictive markers when administering SSAs, especially in GI-NETs. A total of 75% of the Volante score and IRS positive cases presented therapeutic efficacy of SD or CR. However, it is also true that approximately half of the cases determined as the Volante score and IRS negative also demonstrated therapeutic efficacy of SD or CR in this study. Therefore, more objective and rigorous standardization of the SSTR2 immunoreactivity is required as well as possible standardization of pre-analytical, analytical, and post-analytical factors involved in immunohistochemistry [24]. ROC analyses in this study also revealed that DIA could provide a good alternative for predicting response to SSAs in evaluating SSTR2 immunoreactivity of GI-NETs. Eyeball-based analysis had high sensitivity but low specificity (Figure 4); DIA could be more sensitive and more specific, at least compared to the eyeball-based analysis (Figure 4). Potential application of DIA to the evaluation of human epidermal growth factor receptor 2 (HER2) or programmed death ligand-1 (PDL-1) immunoreactivity was reported to provide a more objective and accurate prediction of therapeutic response than that of eyeball examinations [28,34], which was consistent with this study’s results. In addition, evaluation of SSTR2 using DIA may also be useful in predicting the therapeutic efficacy of novel PRRT [21,22] and further studies are warranted to confirm this clinically important aspect.

The limitations of our present study were the relatively small number of midgut NETs and NET G3 cases as well as those receiving SSAs. This is why the AUC value is not so high and the AUCs showed no staristically significant differences. Furthermore, half of Group 3 patients had undergone preoperative therapy. Therefore, further analyses are also required for clarifying the potential value of SSTR2 analysis in predicting the therapeutic responses to SSAs.

## 5. Conclusions

We first demonstrated expression profiles of SSTR2 in GI-NETs among different embryological origins and their correlation with the responses to SSAs. In addition, we also demonstrated the potential usefulness of DIA compared to eyeball-based analysis in evaluating SSTR2 immunoreactivity of GI-NET. This study could lead to selecting the patients with GI-NETs who could benefit from SSA.

## Figures and Tables

**Figure 1 cancers-14-00775-f001:**
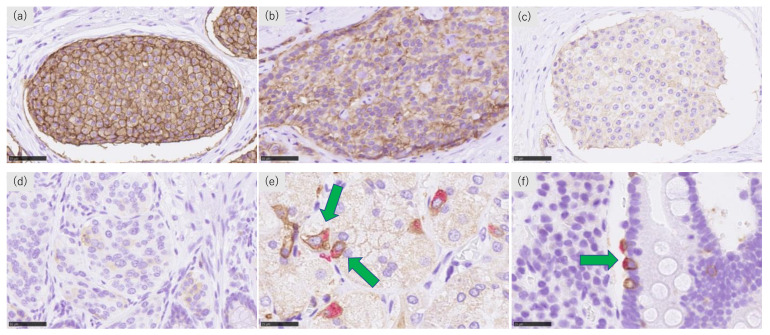
Representative illustrations of SSTR2 and chromogranin A immunohistochemistry. (**a**) Marked and circumferential, (**b**) moderate and incomplete, (**c**) weak and incomplete SSTR2 immunoreactivity in the membrane of tumor cells, (**d**) cytoplasmic SSTR2 immunoreactivity of tumor cells. Double immunostaining of SSTR2 and chromogranin A in the (**e**) stomach and (**f**) duodenum. Green arrows represent SSTR2 and chromogranin A double-positive neuroendocrine cells.

**Figure 2 cancers-14-00775-f002:**
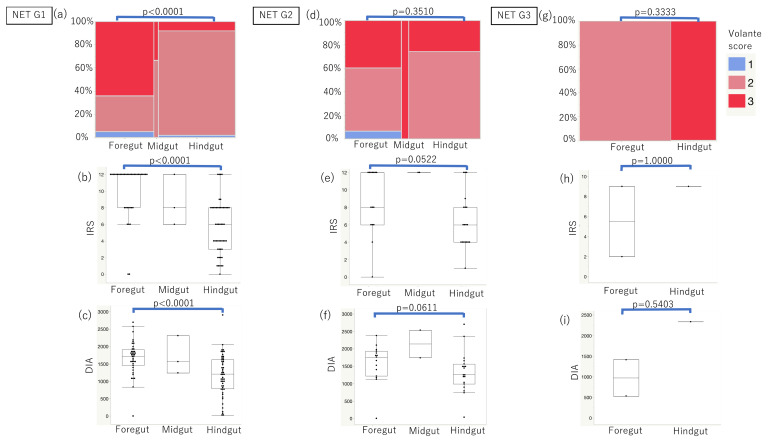
Correlation between SSTR2 immunoreactivity and embryonal sites of origin in NET G1, G2, and G3, respectively. (**a**–**c**) In GI-NETs G1, foregut NETs demonstrated significantly higher immunoreactivity than hindgut NETs according to all scoring systems (*p* < 0.0001). (**d**–**f**) In GI-NETs G2, foregut NETs showed close to significantly higher immunoreactivity than hindgut NETs using IRSs and DIA (*p* = 0.0522, *p* = 0.0611, respectively). (**g**–**i**) In GI-NETs G3, there were no significant differences of SSTR2 immunoreactivity evaluated by all scoring systems between foregut and hindgut GI-NETs.

**Figure 3 cancers-14-00775-f003:**
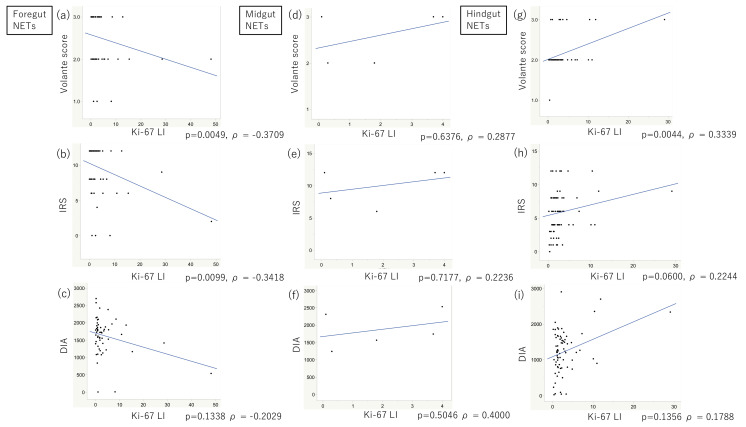
Correlation between SSTR2 immunoreactivity and Ki-67 labeling index (LI). (**a**–**c**) In the foregut NETs, SSTR2 was significantly inversely correlated with Ki-67 LI, especially when using the Volante scores and IRSs (*p* = 0.0049, ρ = −0.3709 and *p* = 0.0099, ρ = −0.3418, respectively). (**d**–**f**) In the midgut NETs, SSTR2 was positively correlated with Ki-67 LI, although not significantly. (**g**–**i**) In the hindgut NETs, SSTR2 was positively correlated with Ki-67 LI, especially when evaluated using the Volante scores and IRSs; the former correlation was significant (*p* = 0.0044, ρ = 0.3339 and *p* = 0.0600, ρ = 0.2244, respectively).

**Figure 4 cancers-14-00775-f004:**
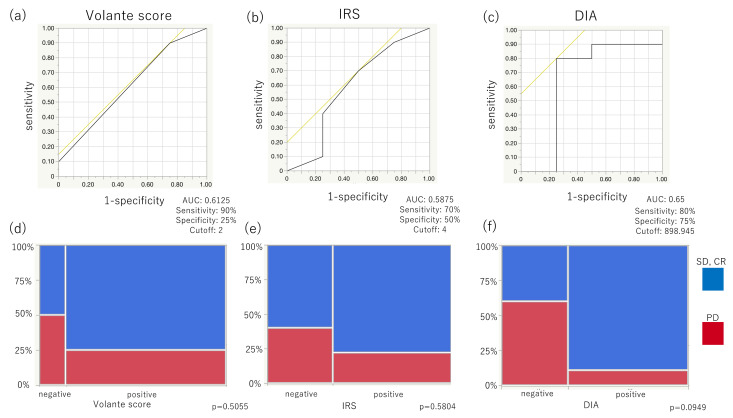
SSTR2 immunoreactivity was evaluated by three different scoring systems and their correlation with the therapeutic efficacy of SSAs in Group 3 patients. (**a**–**c**) The area under the curve (AUC) was highest when evaluated using DIA (AUC, 0.65; cut-off, 898.945; sensitivity, 80%; specificity, 75%), compared with that when using Volante scores (AUC, 0.6125; cut-off, 2; sensitivity, 90%; specificity, 25%) and IRSs (AUC, 0.5875; cutoff, 4; sensitivity, 70%; specificity, 50%). (**d**–**f**) Higher proportions of SD or CR were detected in positive cases than in negative cases regardless of the scoring systems used. (**f**) SSTR2 immunoreactivity evaluated by DIA tended to be closely associated with the therapeutic efficacy of SSAs, but not when evaluated using the Volante scores and IRSs.

**Figure 5 cancers-14-00775-f005:**
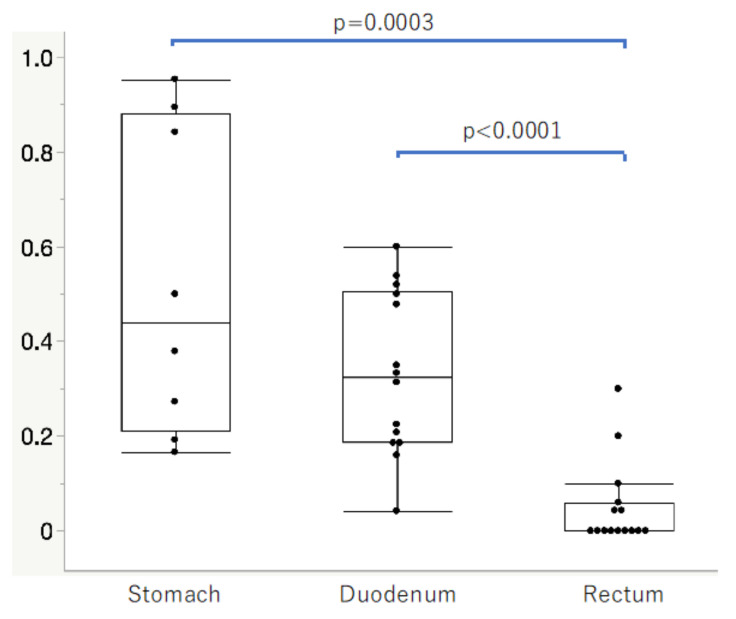
SSTR2-positive rates in the normal neuroendocrine cells of gastrointestinal mucosa in Group 4 cases. SSTR2-positive rates in the stomach and duodenum were significantly higher than those in the rectum (*p* = 0.0003, *p* < 0.0001, respectively).

**Figure 6 cancers-14-00775-f006:**
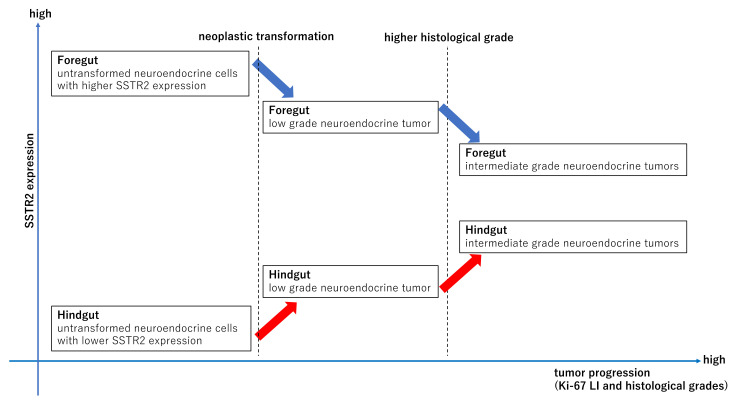
We hypothesized that low grade foregut NETs, derived from their normal counterparts which harbor relatively abundant SSTR2 expression, could have high SSTR2 expression, while foregut NETs with more deviation from their normal counterparts and higher histological grades could have lower SSTR2 expressions. In contrast, hindgut NETs derived from their normal counterparts which harbor intrinsically low SSTR2 expression demonstrated low SSTR2 expression, while hindgut NETs with more deviations from their normal counterparts and higher histological grades had higher SSTR2 expression.

**Table 1 cancers-14-00775-t001:** Summary of the protocols of immunohistochemistry used in this study.

Antibody	Supplier (Catalogue Number)	Host	Antigen Retrieval Treatment	Dilution	Reaction Time of Primary Antibody	Antibody Type	Secondary Antibody	Positive Control
Ki-67	Agilent technologies, US (IR626)	Mouse	PT Link (97 ℃, 20 min), Target Retrieval Solution High PH	Ready to use	20 min, room temperature	MIB-1	EnVision FLEX	Epithelial cell
SSTR2	Abcam, England (ab134152)	Rabbit	AC (121 ℃, 5 min), pH 6.0	1:2000	4 ℃, overnight	UMB1	Histofine Kit	Islet of langerhans
Chromogranin A	Agilent technologies, US (A0430)	Rabbit	Microwave (210 W, 15 min), pH 6.0	1:1500	4 ℃, overnight	polyclonal	Histofine Kit	Islet of langerhans

**Table 2 cancers-14-00775-t002:** Summary of the clinicopathological characteristics of Group 2 patients.

		GI-NET			*p*-Value
		Foregut	Midgut	Hindgut	
Total number		56	5	71	
Age	Median (range)	60.5 (33–88)	62 (53–74)	61 (36–82)	NS
Sex	Male	36	3	46	NS
	Female	20	2	25	
Function	Non-function	42	3	70	<0.0001
	Gastrinoma	13	0	0	
	Carcinoid syndrome	0	2	0	
	Not available	1	0	1	
Hereditary background	Not detected	45	5	71	0.0002
	MEN1	11	0	0	
Volante score	0	0	0	0	<0.0001
	1	3	0	1	
	2	22	2	60	
	3	31	3	10	
IRS	Median (range)	12 (0–12)	12 (6–12)	6 (0–12)	<0.0001
DIA *	Median (range)	1723 (0–2695)	1741 (1240–2533)	1238 (19–2904)	<0.0001
WHO2019	NET G1	38	3	51	NS
	NETG2	15	2	19	
	NET G3	2	0	1	
Ki-67 LI	Median (range)	1.7 (0.1–48.2)	1.8 (0.2–29.1)	1.9 (0.1–29.1)	NS

* Round down after the decimal point. Abbreviations: GI—gastrointestinal; NS—not significant; IRS—immunoreactive score; DIA—digital image analysis; LI—labeling index.

**Table 3 cancers-14-00775-t003:** Summary of the clinicopathological characteristics of Group 3 patients.

Case No.	Sex	Age	Primary Location	Tissue Origin	Grade	Preoperative Therapy	SSTR2 Immunoreactivity			Treatment	Response to SSA	SD, CR = Positive PD = Negative
							Volante Score	IRS	DIA *			
1	M	63	midgut	primary site	G1	TS1, Octreotide	2	4	769	Octreotide, UFT	CR	positive
2	F	60	hindgut	primary site	G2	Not done	2	6	1492	Lanreotide	SD	positive
3	M	55	midgut	primary site	G2	Not done	3	12	1741	Lanreotide	SD	positive
4	M	61	hindgut	primary site	G2	Octreotide	1	0	163	Octreotide	SD	positive
5	F	45	hindgut	primary site	G1	Octreotide, Lanreotide	2	2	1075	Octreotide, Lanreotide	SD	positive
6	M	65	hindgut	primary site	G2	Not done	2	4	898	Octreotide, Lanreotide	SD	positive
7	M	48	hindgut	metastatic site (liver)	G2	Octreotide, Lanreotide	2	2	830	Octreotide, Lanreotide	PD	negative
8	M	72	midgut	metastatic site (liver)	G2	Octreotide, Lanreotide	2	4	731	Octreotide, Lanreotide	PD	negative
9	M	61	foregut	primary site	G2	Not done	2	12	2379	Octreotide	PD	negative
10	M	73	foregut	metastatic site (liver)	G2	Not done	1	0	215	Lanreotide	PD	negative
11	M	74	midgut	primary site	G1	Not done	2	8	1240	Lanreotide	SD	positive
12	F	65	foregut	metastatic site (liver)	G1	Octreotide	2	8	1467	Octreotide	SD	positive
13	F	67	midgut	primary site	G1	Octreotide, Everolimus	2	2	967	Octreotide	SD	positive
14	F	71	hindgut	primary site	G2	Not done	2	4	1027	5FU, CDDP, Octreotide	SD	positive

* Round down after the decimal point. Abbreviations: IRS—immunoreactive score; DIA—digital image analysis; SSA—somatostatin analogue; UFT—uracil-tegafur; 5FU—5-fluorouracile; CDDP—cisplatin; SD—stable disease; CR—complete response; PD—progressive disease.

**Table 4 cancers-14-00775-t004:** Summary of the clinicopathological characteristics of Group 4.

Case No.	Site		Age	Sex	Positive Rates *
1	Stomach	Pyloric gland	64	Male	0.953
2		Pyloric gland	81	Male	0.842
3		Fundic glands	55	Male	0.5
4		Pyloric gland	83	Female	0.166
5		Cardiac gland	83	Male	0.379
6		Fundic glands	66	Male	0.894
7		Fundic glands	63	Male	0.192
8		Cardiac gland	58	Female	0.272
1	Duodenum	Bulb of duodenum	64	Male	0.52
2		Bulb of duodenum	81	Male	0.16
4		Bulb of duodenum	83	Female	0.333
5		Bulb of duodenum	83	Male	0.208
8		Bulb of duodenum	58	Female	0.6
9		Bulb of duodenum	73	Male	0.538
10		Bulb of duodenum	69	Female	0.5
11		Second portion	67	Male	0.187
12		Second portion	71	Female	0.041
13		Second portion	79	Male	0.218
14		Second portion	55	Male	0.185
15		Second portion	67	Male	0.346
16		Second portion	69	Male	0.478
17		Second portion	73	Male	0.313
18	Rectum		60	Female	0
19			83	Female	0.3
20			52	Male	0
21			70	Female	0
22			40	Male	0
23			62	Male	0.1
24			51	Female	0
25			81	Female	0.2
26			75	Male	0
27			70	Female	0.043
28			56	Female	0.043
29			87	Female	0
30			79	Female	0
31			77	Male	0.058
32			37	Male	0

* Round down to the fourth decimal place.

## Data Availability

The data in this study are available from the corresponding authors. The data are not publicly available because of ethical restrictions.

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
