# Peer review of "Somatostatin Receptor 2 Expression Profiles and Their Correlation with the Efficacy of Somatostatin Analogues in Gastrointestinal Neuroendocrine Tumors"

_cancers, 2022, doi:10.3390/cancers14030775_

Round 1
Reviewer 1 Report
The objective of this project was to correlate SSTR2 expression profiles and correlation with the efficacy of SSAs.
- Immunoreactivity and embryologic origin and proliferative activity
- Therapeutic efficacy
- Comparison to non tumor tissue.
SSTR2 assessment utilized eye assessment with (1) the Volante score and (2) Immunoreactivity score; and to eliminate the intra-observer variability, digital image analysis (DIA) was employed.
Sample size was 145 GI-NET specimens from 141 patients
Excluded patients who received chemotherapy prior to surgery and those with metastatic lesions, leaving 132 specimens for analysis.
14 specimens from patients who had received SSA
37 non-neoplastic mucosa from 32 of the patients. No h/o NET or CTX
- this was a reasonable sample size, though small cohorts of treated and mid gut which limits extrapolation of the studies results
Methods were clear.
Adequate statistical analysis methods
Results
Eyeball methods had high sensitivity but low specificity. Whereas DIA was more sensitive and specific.
The data highlights the limitations of immunoreactivity assessments, and agree that other more objective methods are needed. This is a good start. A larger analysis addressing the limited samples of patients with midgut, and those treated with SSA only would be the next step.
Author Response
Response to Reviewer 1 Comments
The objective of this project was to correlate SSTR2 expression profiles and correlation with the efficacy of SSAs.
- Immunoreactivity and embryologic origin and proliferative activity
- Therapeutic efficacy
- Comparison to non tumor tissue.
SSTR2 assessment utilized eye assessment with (1) the Volante score and (2) Immunoreactivity score; and to eliminate the intra-observer variability, digital image analysis (DIA) was employed.
Sample size was 145 GI-NET specimens from 141 patients
Excluded patients who received chemotherapy prior to surgery and those with metastatic lesions, leaving 132 specimens for analysis.
14 specimens from patients who had received SSA
37 non-neoplastic mucosa from 32 of the patients. No h/o NET or CTX
- this was a reasonable sample size, though small cohorts of treated and mid gut which limits extrapolation of the studies results
Methods were clear.
Adequate statistical analysis methods
Results
Eyeball methods had high sensitivity but low specificity. Whereas DIA was more sensitive and specific.
The data highlights the limitations of immunoreactivity assessments, and agree that other more objective methods are needed. This is a good start. A larger analysis addressing the limited samples of patients with midgut, and those treated with SSA only would be the next step.
Author’s response: We agreed with your comments. We therefore emphasized the potential importance of exploring SSTR2 status in a larger study including more cases of midgut NETs and GI-NET cases treated with SSA and revised the manuscript accordingly.
Reviewer 2 Report
Watanabe et al. analyse the expression of somatostatin receptor type 2 in neuroendocrine tumours of the gastrointestinal tract, and correlate their results with a subset of patients, who received somatostatin analogue therapy. Besides eyeball-based evaluation of immunohistological staining, the authors also investigated suitability of digital image analysis (DIA). The results presented are interesting, but presentation and clarity should be improved before publication. I would advise to seek help from a native English speaker, since some sentences are written very oddly, partly with wrong word order making it hard to follow. One of many examples is the sentence on line 65-67. Please address the following major points:
- The authors compare different methods for scoring IHC results. Please also provide a correlation analysis between the methods in the supplement. Do they generally agree with each other? Additionally, please comment on how similar the 2 investigators scored in the eyeball-based methods. What happened in case of disagreement?
- The AUC for DIA was calculated at 0.65 compared to 0.59 and 0.61 for the other methods. I suspect that there is no statistical difference. Could you please check? Please put these AUC numbers into context in your discussion. Such an AUC is quite poor and would have low diagnostic benefit. A much higher number of patients is required to make any statement on the whether DIA performs better than the other scoring methods. You tried to address this in the limitations, but it requires improvement. Based on these criticisms, please rephrase the statement on line 51-52.
- In the methods, please add the catalogue numbers or clone IDs for the different antibodies. How did you validate the antibodies? Did you use isotype controls?
- Figure 6 is a nice idea and should definitely be included; however, I find the execution quite poorly. What do the blue arrows mean and what do they add to the message? In addition, the DNA damage symbol looks out of place, I would remove it. Change "deviation from normal neuroendocrine cells" to tumor progression. Why did you write immature neuroendocrine cells? "untransformed"neuroendocrine cells" might be better. Connected to this figure, there is a confusing statement in the discussion (line 365-67): “more deviation from normal counterparts in neoplastic transformation”. Please improve the figure and the discussion connected to it.
- Please also mention in the introduction or discussion that SSTR2 is the target of peptide receptor radiotherapy using 177Lu-DOTATATE.
- Line 72-73: Is there a reason for the apparent increase in GI-NET incidence?
Minor points:
- Line 225: incorrect reference to Figure S1, it should be S3.
- In Table 2 and 3, please reduce the number of digits after the comma for DIA to max. 1 (even better none). I cannot imagine that 3 or more digits after the comma add anything.
- Table 4: Reduce digits after the comma to max. 3 for positive rates.
Author Response
Response to Reviewer 2 Comments
Point 1. The authors compare different methods for scoring IHC results. Please also provide a correlation analysis between the methods in the supplement. Do they generally agree with each other? Additionally, please comment on how similar the 2 investigators scored in the eyeball-based methods. What happened in case of disagreement?
Author’s response 1: We agreed with your comments. Therefore, we evaluated the correlation of the findings obtained among the different methods employed. Results did demonstrate the marked correlation among the three different methods of evaluation (lines 264-268). When disagreement occurred in the eyeball-based methods between these two observers, consensus was obtained by simultaneous observation using a dual-headed microscope with discussion (lines 197-199). We revised the manuscript accordingly.
Point 2. The AUC for DIA was calculated at 0.65 compared to 0.59 and 0.61 for the other methods. I suspect that there is no statistical difference. Could you please check? Please put these AUC numbers into context in your discussion. Such an AUC is quite poor and would have low diagnostic benefit. A much higher number of patients is required to make any statement on the whether DIA performs better than the other scoring methods. You tried to address this in the limitations, but it requires improvement. Based on these criticisms, please rephrase the statement on line 51-52.
Author’s response 2: We agreed with your comments above. AUCs were evaluated by the DeLong test. The significance level of the DeLong test was corrected using Bonferroni's inequality (lines 258-260). No statistically significant differences were detected in those AUCs (lines 332-333). We revised the manuscript accordingly.
Point 3 In the methods, please add the catalogue numbers or clone IDs for the different antibodies. How did you validate the antibodies? Did you use isotype controls?
Author’s response 3: We agreed with your comments above. Therefore, we added the catalogue numbers or clone IDs of the different antibodies (Table 1) and revised the manuscript accordingly. We validate the specificity of antibodies by employing representative positive controls of immunostaining . The antibody, UMB1, was a validated antibody, used in many reported studies and membranous staining obtained by immunohistochemistry was also reported to be correlated with the result of somatostatin scintigraphy and effect of somatostatin analogues [18,19]. We revised the manuscript accordingly.
Point 4. Figure 6 is a nice idea and should definitely be included; however, I find the execution quite poorly. What do the blue arrows mean and what do they add to the message? In addition, the DNA damage symbol looks out of place, I would remove it. Change "deviation from normal neuroendocrine cells" to tumor progression. Why did you write immature neuroendocrine cells? "untransformed neuroendocrine cells" might be better. Connected to this figure, there is a confusing statement in the discussion (lines 365-67): “more deviation from normal counterparts in neoplastic transformation”. Please improve the figure and the discussion connected to it.
Author’s response 4: We agreed with your comments above. Left blue arrow indicated neoplastic transformation and right blue arrow indicated higher histological grades. SSTR2 profiles in GI-NETs were reasonably postulated to be different according to their primary sites. Therefore, the difference of SSTR2 profiles between foregut and hindgut NETs were also considered to be derived from the SSTR2 status of normal neuroendocrine counterpart cells. We modified Figure 6 and the discussion (lines 372-398) accordingly.
Point 5. Please also mention in the introduction or discussion that SSTR2 is the target of peptide receptor radiotherapy using 177Lu-DOTATATE.
Author’s response 5: We appreciate your comments and added the sentence regarding the peptide receptor radiotherapy using 177Lu-DOTATATE in the Introduction and Discussion (lines 94-96 and 417-419) and revised the manuscript accordingly.
Point 6. Line 72-73: Is there a reason for the apparent increase in GI-NET incidence?
Author’s response 6: The incidence of gastrointestinal neuroendocrine tumors (GI-NETs) has recently increased, possibly due to increased administration of proton pump inhibitors and development of detection equipment such as endoscopy [1–2][3] (lines 75-76).
Minor points:
Point 1. Line 225: incorrect reference to Figure S1, it should be S3.
Author’s response 1: We appreciate your comments. We modified the manuscript accordingly (line 273).
Point 2. In Table 2 and 3, please reduce the number of digits after the comma for DIA to max. 1 (even better none). I cannot imagine that 3 or more digits after the comma add anything.
Author’s response 2: We appreciate your comments above. We modified the manuscript accordingly (Tables 2, 3).
Point 3. Table 4: Reduce digits after the comma to max. 3 for positive rates.
Author’s response 3: We appreciate your comment. We modified the manuscript accordingly (Table 2, 3).
Round 2
Reviewer 2 Report
- Line 94-6: Reference 21 seems not quite fitting with the general statement about PRRT. It might be better to refer to a recent review on the matter.
- Line 409-11: Please change this statement: The comparison of the AUCs yielded no difference, this means, you cannot say that DIA is more accurate. Maybe you could speak about a trend towards higher specificity. Please change all sentences with the wording “efficacy predictability of SSAs” in the summary and abstract. You could say: DIA provides a good alternative for predicting response to SSAs.
Author Response
Response to Reviewer 2 Comments
Point 1. Line 94-6: Reference 21 seems not quite fitting with the general statement about PRRT. It might be better to refer to a recent review on the matter.
Author’s response 1: We agreed with your comments and added recent review article of PRRT in reference [22]. (line 95, 425, 547)
Point 2. Line 409-11: Please change this statement: The comparison of the AUCs yielded no difference, this means, you cannot say that DIA is more accurate. Maybe you could speak about a trend towards higher specificity. Please change all sentences with the wording “efficacy predictability of SSAs” in the summary and abstract. You could say: DIA provides a good alternative for predicting response to SSAs. The authors compare different methods for scoring IHC results. Please also provide a correlation analysis between the methods in the supplement. Do they generally agree with each other? Additionally, please comment on how similar the 2 investigators scored in the eyeball-based methods. What happened in case of disagreement?
Author’s response 1: We agreed with your comments and revised the manuscript accordingly. (line 51-52, 67-68, 417-418)